# The Impact of Open Access on Teaching—How Far Have We Come?

**Elizabeth Gadd** [1,*], **Chris Morrison** [2]  **and Jane Secker** [3]

1   Research Office, Loughborough University, Loughborough LE11 3TU, UK
2   Information Services, University of Kent, Templeman Library, Canterbury, Kent CT2 7NU, UK
3   LEaD, City, University of London, Northampton Square, London EC1A 0HB, UK
*   Correspondence: e.a.gadd@lboro.ac.uk

**Abstract:** This article seeks to understand how far the United Kingdom higher education (UK HE) sector has progressed towards open access (OA) availability of the scholarly literature it requires to support courses of study. It uses Google Scholar, Unpaywall and Open Access Button to identify OA copies of a random sample of articles copied under the Copyright Licensing Agency (CLA) HE Licence to support teaching. The quantitative data analysis is combined with interviews of, and a workshop with, HE practitioners to investigate four research questions. Firstly, what is the nature of the content being used to support courses of study? Secondly, do UK HE establishments regularly incorporate searches for open access availability into their acquisition processes to support teaching? Thirdly, what proportion of content used under the CLA Licence is also available on open access and appropriately licenced? Finally, what percentage of content used by UK HEIs under the CLA Licence is written by academics and thus has the potential for being made open access had there been support in place to enable this? Key findings include the fact that no interviewees incorporated OA searches into their acquisitions processes. Overall, 38% of articles required to support teaching were available as OA in some form but only 7% had a findable re-use licence; just 3% had licences that specifically permitted inclusion in an 'electronic course-pack'. Eighty-nine percent of journal content was written by academics (34% by UK-based academics). Of these, 58% were written since 2000 and thus could arguably have been made available openly had academics been supported to do so.

**Keywords:** open access; education; teaching support; licensing

## 1. Introduction

One of the side-effects of academic authors assigning copyright in their scholarly outputs to publishers is that academics then lose the right to re-use that content in their own teaching and to permit others to do the same. Whilst in some cases authors may receive permission from the publisher to re-use their content in their own teaching [1], it is unlikely that they will receive permission to extend that right to others. Some of the early Jisc eLib projects demonstrated that clearing copyright permissions was one of the main barriers to creating electronic access to such content for teaching [2,3]. Most journal content is now 'born digital' and re-use for teaching purposes is often possible through e-journal licences. For other content, such as journals not owned by an institution, or books, the Copyright Licensing Agency's (CLA) higher education (HE) licence is available. CLA are a UK collective management organisation (CMO) who offer blanket licences to UK universities (and other organisations) to allow copying of book and journal content beyond what copyright exceptions would permit. The licence fees are then distributed as royalty payments to authors, publishers and artists in recompense for potential loss of sales. The CLA HE licence [4] allows higher education institutions (HEIs) to make multiple copies (either scanned or photocopied) of publisher-owned content (up to 10% of a work or

one chapter from a book/one article from a journal–whichever is the greater) available under certain conditions for cohorts of students. This costs the UK HE sector £15.54 M per annum [5] on top of the annual subscription fees they already pay to journal publishers, and the article processing charges (APCs) they may pay to make content available on Gold open access [6].

One of the perceived benefits of the open access movement was that it would not only enable researchers to quickly and easily access content to support their research, but it would also enable libraries (on behalf of academics) to provide digital access to content to support the teaching of cohorts of students, without the need to clear individual permissions or to pay additional licence fees. Of course, the open access movement has, to date, focussed on the 'royalty-free' literature such as journal articles and conference papers, rather than literature that usually attracts royalties such as monographs. However, Research England has already indicated that they expect any monograph submissions to the 2027 Research Excellence Framework (REF) to be available on open access [7]. Whilst such monographs are likely to be research monographs and not textbooks, the latter of which are more likely to be in demand for supporting teaching, this policy is still likely to have an impact on the open availability of book content to support teaching in the future. In addition around the world, particularly in North America but also a number of other countries, the open textbook movement is starting to make significant progress, and is leading to a reduction in the costs paid by students and institutions when providing access to teaching materials. Despite only 15 UK institutions using open textbooks the current savings to students have been estimated at $1 million (or almost £800,000) [8].

This study seeks to understand how far the UK HE sector has progressed towards open access (OA) availability to the scholarly literature it requires to support courses of study. It uses data submitted to the CLA on items scanned and copied under the CLA HE licence, a series of interviews with users of the licence, and the results of a workshop with acquisitions librarians, to answer the following research questions:

*Research Questions*

RQ1: What is the nature (type and age) of the content being scanned or copied from digital originals to support courses of study?

RQ2: Do UK HE establishments regularly incorporate searches for open access availability into their acquisition processes to support courses of study? If not, why not?

RQ3: What proportion of content used under the CLA Licence is already available on open access, and does it have an appropriate re-use licence?

RQ4: What percentage of content used by UK HEIs under the CLA Licence is written by academics (either UK-based or non-UK-based) and thus has the potential for being made open access had those HEIs had policies to support their academics to enable this?

By utilising the open access content sourcing services OA Button [9] and Unpaywall [10] to assess the availability of items copied under the CLA Licence, we are able to provide a comparative review of the performance of these services and the extent they can be relied upon to find OA versions of 'real world' content in demand by UK HEIs to support courses of study.

## 2. Literature Review

Open access to the "royalty-free" literature [11] was promoted as a solution to two related problems facing the world of scholarly communication: access and impact [12]. The access problem was the result of spiralling journal subscription costs, leading to cancelled library subscriptions. Scholars as readers could no longer access all the literature they needed, and consequently could not read and cite relevant papers. This led to the impact problem: scholars as authors have unread and uncited papers, thus reducing their impact. Open access was given a firm foundation in 2002 through the Budapest Open Archive Initiative (BOAI) Declaration [13]. The Declaration provided the first community-agreed definition of open access and was closely followed by the Berlin declaration and Bethesda definition, both in 2003. The key elements of these definitions were similar, which led Suber to refer to all three as

the "BBB definition" [14]. The common elements were that the results of scholarly enquiry should be freely available to anyone with an internet connection, and subject to permission-free scholarly re-use. It is the latter element that was of particular interest to the open education resources (OER) movement [15] which grew up at about the same time. As Pinfield and Corrall note, "for OERs and MOOCs [Massive Open Online Courses] to achieve their full potential they often require other complementary Opens, including open textbooks and research outputs [16]." Such open content was also identified as a potential solution to the problem of providing access to course readings for large groups of undergraduate students [17].

The promise of digitally available content to help universities provide access to large cohorts of students has been well-studied [18]. Research undertaken in 2015 showed that the growth in scanning activities under the CLA HE Licence, had risen from a median of 300 scans per institution in 2009 to 940 in just four years [19]. In the same period there has been a significant take up of online reading list services such as Talis Aspire, and the CLA HE Licence with its Digital Content Store facility, as testament to the demand for this [20]. However, to date, there appears to be little engagement with OA content as a means to meet the needs of growing student numbers. This may be due to the differing needs of academics-as-educators who seek to use OA content in teaching compared to the needs of academics-as-researchers who seek to use OA content to support their research. Some of the key characteristics of OA content that may have an impact on its use in teaching are outlined below.

**Gratis vs libre.** Suber [21] made the early distinction between *gratis* OA where content is available free of charge, and *libre* OA where content is available free of usage restrictions. The difference between the two is both technological (libre OA content may be available in a form that can be text and data-mined), and legal (libre content will be available under a liberal re-use licence). Whilst many researchers' needs may be met through gratis OA (unless they wish to undertake text and data mining), teachers wishing to incorporate OA content into anthologies or other teaching collections may benefit more from libre OA. However, research by Harold and Rolfe [22] suggests that academic staff understanding about open education initiatives is still relatively limited and in a survey of 45 academics it was reported that respondents found it "hard to tell what different licences for content mean, in terms of how they can be used".

**Gold vs Green.** Research by Fry et al. has shown that a researcher's discipline has a bearing on whether and how they engage with different forms of OA [23]. In some disciplines (e.g., Physics) researchers may be equally served by reading a 'green' self-archived author manuscript as a 'gold' publisher-provided PDF. However, whilst there is currently no evidence as to teachers' preferences, it might be hypothesised that, irrespective of discipline, they would prefer to provide their students with access to gold publisher PDFs on information literacy grounds. For example, teaching students about the legitimacy of content they find on the internet may involve looking for markers of provenance, such as publisher name, also, the use of paginated publisher PDFs may encourage students to cite properly. Gold publisher PDFs are also more straightforward for students to find through library search engines. There is also an increasing recognition of the need to teach what Grgic [24] calls 'open access literacy'. This should no doubt include the increasing prevalence of 'bronze' OA where a publisher may make a copy of an articled available for a period of time at their discretion and withdraw it without warning—clearly a form of OA that educators cannot rely on.

**Legal vs illegal.** Jamali found that 51.3% of articles found on ResearchGate were illegally mounted publisher PDFs [25]. The legality around drawing others' attention to, or reading, an illegally mounted journal article is somewhat unclear given recent case law considering the scope of "communication to the public" under EU law [26]. However, that aside it is known that Librarians, due to their inherent professional conservatism, are particularly risk-averse [27] and are unlikely to sanction access to content of dubious origin.

**Permanence vs impermanence.** One of the main reasons Librarians are unlikely to want to point students towards illegal OA copies is because of their potential instability. There have been a number of legal actions against ResearchGate [28] and sites such as SciHub [29] by publishers resulting in the

removal (or impermanence) of illegally mounted content. Piwowar and Priem's study of OA content identified by the Unpaywall service found that 15.3% of items were so-called 'bronze' OA which may or may not remain permanently available [30]. Work by Bjork and colleagues discovered that the permanence of green OA copies was also poor if not made available via a managed repository. They found "the persistence of green OA copies was lowest on arbitrary websites, such as personal or departmental websites, where the items could be found [three years later] in only 56% of cases" [31]. To be of benefit to a researcher, an article only needs to be available on the day they search for it. At that point they can download it thus making it permanently accessible to them. However, any links to OA content made available to students need to be reliable so that a cohort of students may access them over the course of the study period.

**Age of content.** Open access is a twenty-first century phenomenon. In general, researchers in journal-based disciplines need to access the most recent research and so OA serves them well. Indeed, Piwowar and Priem found that 47% of the scholarly journal literature being sought by researchers via the Unpaywall application was available to them as OA, despite a much lower estimate (28%) being available as OA overall [30]. Martin et al., discovered that 54.6% of items with a publication date of 2009 or 2014 on Web of Science were also available on Google Scholar [32]. However, the resources used to support teaching are not all from the twenty-first century. There is not a great deal of current literature on the age of reading list material used in support of teaching, however, an analysis of reading list items performed on the Jisc ACORN Project back in 1998 showed that some requested articles were fifty years old, and 35% were over eight years old [33]. It would therefore seem like a fair assumption that students require greater access to older content than researchers, something this study seeks to investigate.

**Geographical origin of content.** In a globally connected world with an increasing volume of co-authorships it might seem artificial to make reference to the geographical origins of content. However, in OA terms, due to the influence of national OA policies and legislation, the availability of OA content does vary significantly by region. Work by Jubb et al., showed that as a result of the OA policy approaches taken in the UK, the proportion of OA journal articles in 2016 accessible immediately on publication was 37% relative to a proportion of 24% globally [34]. Gadd, Fry and Creaser also found that UK and US publishers' OA policies appeared to be influenced by the national OA policy environment [35]. It therefore seems fair to say that if teachers in a particular country tend to use content from that same country more frequently in their teaching (something explored by this study) then the national OA policy framework will have a considerable bearing on the success of that activity.

**Type of content.** Due to the non-royalty-based nature of the scholarly research literature, much of the focus of the OA movement to date has been on journal articles rather than books. Whilst it would seem logical that the research article is likely to be of more relevance to the researcher than the student, and conversely, the textbook is likely to be more relevant to the student than to the researcher, there is scarce evidence in the literature to support either of these assertions. A study of academic reading at Loughborough University showed that students self-reported making the most use of websites (69%) followed by books (60%) to support their studies. However, "the use of journals increased as students progressed from year to year, with just 16% (47) of first years using them frequently. These figures rose to 41% (106) of third years and 53% (43) of fourth years [36]." A study of students' use of the research literature by Jisc showed that "most students use research to support their assignments, so use of research is primarily 'assessment led'" [37]. However, this seemed to be changing with "some students demonstrat[ing] a sophisticated engagement with research which they use to develop arguments rather than simply support a point". There have been anecdotal observations of an increased engagement with research articles over long-form monographs as a result of the more manageable length of an article being a more attractive and consumable size for the busy undergraduate. This is likely to vary by discipline. Brewerton has demonstrated the differing lengths of reading lists by discipline [38,39]. It may even vary by institution as they seek to support learners with different backgrounds.

**Summary.** Although the impact of open access on teaching is yet to be studied in any depth, through what we know about the approaches of HE teachers, the needs of undergraduate students, and the characteristics of OA content, we can make some assumptions about the suitability of OA content to support HE teaching. This study provides further insight into the nature of the material required to support teaching and the availability of suitable OA content to meet those needs.

## 3. Methodology

This study took a mixed methods approach, using interviews to form case studies, a workshop and a quantitative data analysis.

### 3.1. Structured Interviews

Structured interviews were part of a case study methodology, based on a purposeful sample of ten higher education institutions, to understand the decision-making processes that inform both the purchasing of printed and electronic resources to support teaching and the use of the CLA Licence. In all cases the interviewees were the acquisitions librarians or the librarian who coordinated the reading list service. In three instances two people took part in the interview as the processes overlapped with their responsibilities. One copyright specialist participated in an interview with their acquisitions librarian. The interviews all took place in October 2018 and a list of the questions is included in Appendix A. As the questions largely involved providing factual information or describing processes rather than asking their opinions, the decision was taken not to transcribe the interviews but to make detailed notes. Therefore direct quotations are not available. The interviews were all confidential and interview transcripts were anonymised to protect the identify of the 10 institutions.

Institutions were selected based on the overall spend on information provision and their use of the CLA Licence. The CLA dataset and the Society of College and National University Libraries (SCONUL) Statistical Return was used to help select the cases. The sample included:

- Four institutions who reported high use of the CLA Licence and had a high spend overall on resource provision. All four institutions were Russell Group institutions with student numbers in excess of 25,000.
- Five institutions who reported low use of the CLA Licence and had a lower overall spend on resource provision. All institutions were post-92 universities or specialist colleges in the arts with fewer than 20,000 students.
- One institution who reported high spend on resource provision and low use of the CLA Licence. This was a Russell Group institution with fewer than 25,000 students.
- Data from these institutions were collected by telephone and face-to-face interviews and followed up with email correspondence to ensure the data collection exercise had been accurate.

### 3.2. Workshop with Acquisition Librarians

A second data collection exercise took place at the National Acquisitions Group (NAG) forum in November 2018, with a larger group of institutions who were asked a subset of the interview questions to triangulate with the data collected in the case studies. The NAG forum is primarily attended by acquisitions librarians who manage subscriptions, purchase books and journals and often manage reading list services. During the workshop the delegates were asked a series of questions and notes were written up by a nominated participant on each table. Some of the question responses were also collected via a polling system used at the conference and these helped to sense check the data collecting as part of the case studies. The questions asked in the workshop are included in Appendix B. The group were also presented with some interim findings from the 10 case studies and given the opportunity to discuss the data and compare them to their own experiences.

### 3.3. CLA Data Analysis

In order to distribute revenues from the CLA HE Licence to rightsholders, the CLA collect data from HEIs as to which items they are copying in digital format. This 'data reporting' feature where bibliographic data are submitted annually to CLA was introduced in 2006, partly to allow rightsholders to monitor the impact of scanning and digital-to-digital copying. Copying under the CLA Licence can take one of three forms: print-to-print (photocopying), print-to-digital (scanning) and digital-to-digital (digital copying). This analysis uses the scanning and digital copying data reported to the CLA during 2016–2017 under the terms of a data sharing protocol agreed and expressed in clause 7.4 of the HE Licence [40]. A total of 209,512 scanning records and 13,185 digitally copied items formed the foundation for this analysis. For the most part, each record contained only a manually entered title, page range and international standard number (ISN). Items were from a wide range of disciplines predominantly from Arts and Humanities, Social Sciences, Nursing and BioSciences.

Of the 209,512 records on the 'scanning reported' spreadsheet, a large proportion (19%) had no identifiable ISSN or ISBN (i.e., it had 5/6/7 characters). As an identifiable ISN was important for locating the item in question, those without one were excluded from the study, this left 169,900 records. Of the 13,185 records on the 'digital reported' spreadsheet a much higher proportion, 13,103 (99%) had an identifiable ISBN or ISSN.

### 3.4. Sample Selection

A random stratified sample was taken. To achieve a confidence level of 95% at a confidence interval of +/− 5, a sample of 250 journals and 378 books were needed from the scanning reported sheet and 130 journals and 5 books were needed from the digital reported spreadsheet (see Table 1).

**Table 1.** Identifiable international standard numbers (ISNs) (required sample size).

|  | Books | Journal Articles | Sampled Journal Articles Available | Total |
|---|---|---|---|---|
| Print to Digital (Scanned) | 148,249 (378) | 21,651 (250) | 188 | 169,900 |
| Digital to Digital (Digitally Copied) | 1865 (5) | 11,238 (130) | 108 | 13,103 |
| Total | 150,114 (383) | 32,889 (380) | 296 | 183,003 |

Unfortunately, only nineteen of the 59 institutions (32%) supplying data provided author information in their submission to the CLA, meaning that 17,034 of the 21,651 scanning records (78%), and 7650 of the 11,238 digitally copied records (68%) with an associated ISSN did not contain any author information. Those records without author information only had journal name and page numbers which was not enough to locate the full-text and ascertain its availability. The sample therefore had to be taken from the 4617 scanning records and 3588 digitally copied records, containing author information. This is clearly something of a limitation of the research and it is recommended that arrangements are made in future for the full records of copied content to be captured by reporting HEIs and made available to the community for future study.

Of the 250 scanned journal articles, bibliographic data could only be ascertained for 196 (78%) of them based on the author, journal, and page numbers provided. For a further eight of the scanned items, only the bibliographic reference could be found and no other availability data, leaving 188 usable records. Of the 130 digital items, bibliographic data was found for 108 (83%) of them. This reduced the confidence interval of our data from +/− 5 to +/− 6 at the 95% confidence level.

A small group of Librarians working to an agreed and piloted methodology ran searches using Google Scholar [41], and/or WorldCAT [42] to supplement the CLA data with full bibliographic details and where possible, the DOI of each item in the sample. The results were compiled in a shared Excel spreadsheet. Using affiliation data on the publication itself, it was noted whether (at the time of writing) the author was from a UK or non-UK academic institution. To ascertain whether an item

was on open access, two freely available tools were used in addition to Google Scholar: the Open Access Button (OAB) [9] and the Unpaywall [10] plug in. The locations of the copies found by all three services were noted.

The OAB was launched in 2013, supported by SPARC (the Scholarly Publishing and Academic Resources Coalition) founded by the American Research Librarians (ARL) group. It is a website and app which searches thousands of open access sources such as repositories and publisher websites, for open access versions of articles. End-users may search for OA content via OAB using unique IDs such as a DOI or PubMed ID, or even a bibliographic citation or URL. If an OA version is not found, an option is given for the requester to submit an email request to the author of the article—although the requester may have to supply the email address of the author if OAB is not able to identify it.

Unpaywall was launched in 2016 by ImpactStory founders Heather Piwowar and Jason Priem [43], although its underpinning technology, oaDOI, was developed much earlier. It consists of a database of over 20 million freely available scholarly articles and a browser plugin that uses the oaDOI technology to identify if an open access version is available of the article currently displaying in the user's web browser. It also highlights if the found version is Gold, Green or Bronze OA. Neither Unpaywall nor OAB index content available in Academic Social Networking sites such as Academia.edu and ResearchGate.

If an OA version was found, it was noted whether any human-readable licence terms were attached to it. If so, they were assessed to see if they gave permission for HE libraries to make the content available in an electronic coursepack. This is not the only educational use to which such content might be put by HEIs, however, it is the main purpose of those scanning content under the CLA Licence and thus made a good test case.

## 4. Findings

### 4.1. RQ1: What Is the Nature (Type and Age) of the Content Being Scanned/Copied from Digital to Support Courses of Study?

It was interesting to note that of the 169,900 scanning records, the vast majority 148,249 (87%) were books and only a minority, 21,651 (13%) were journals. However, of the 13,103 digital copying records, the same proportions were observed but in reverse. Namely, 11,238 were journals (86%) and 1865 (14%) were books. Taken together, 18% of the digital content made available under the CLA Licence during 2016–2017 was from journals and 82% from books.

Of course, the volume of journal items copied under the CLA HE Licence should not necessarily be seen as a reflection of the low levels of journal content being used to support HE courses of study due to the fact that much journal content could be legitimately re-used for teaching under the original e-journal licence. Indeed, in interview, it became clear that institutions would check whether journal articles were available in digital format, through their current subscriptions and then whether it could be purchased in born digital format, before relying on the CLA Licence.

As the focus of this article is on open access availability of teaching content, and the majority of content available as OA is journal articles, the remainder of this article relates to journal articles.

Publication Dates of Journal Articles Copied under the CLA Licence

The publication dates of journal articles copied from digital originals ranged across 48 years from 1969–2017 with the average (mean) year of an item from the digital sample being 2006 and a mode of 2014. By contrast, the publication dates of scanned journal articles ranged across 98 years from 1918–2016, with the mean and modal publication date being 1995. Figure 1 shows the spread of publication dates of articles either copied from a digital original or scanned from a print original.

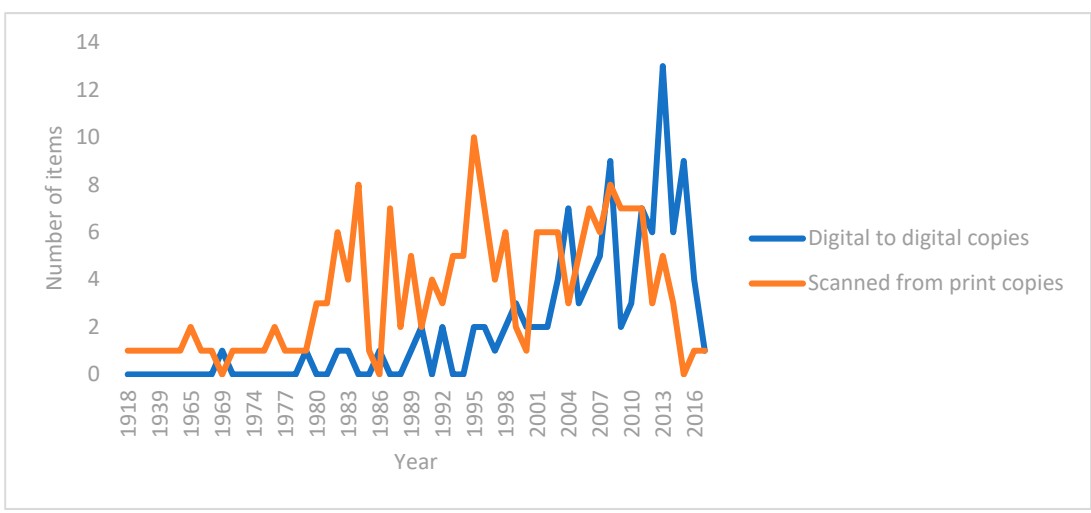

**Figure 1.** Publication dates of scanned/copied journal articles.

*4.2. RQ2: Do UK HE Establishments Regularly Incorporate Searches for Open Access Availability into Their Acquisition Processes to Support Individual Courses of Study? If Not, Why Not?*

During the interviews none of the ten institutions reported that they currently check for open access availability when looking to acquire content for teaching purposes. One institution said they were about to introduce this into their processes, and two said that the discovery tool the acquisitions staff used sometimes flagged up open access content. However, most people felt the time it would take to introduce this procedure might not be worth the effort. One institution also felt some clear guidance on where to search, developed either at a national level, or with help from the research support team in their library would greatly assist in this process. Two institutions felt that it was difficult to ascertain if the OA content could be re-used under a licence or if it was available legitimately which discouraged them from using it for teaching purposes.

Furthermore, asking this question led most respondents to question how likely open access content might be found, as it was clearly not something that had been considered in any great detail. A similar finding was reported in the NAG workshop in Table 2, where only one institution said they would check for open access readings as a matter of course, but all said if they knew it was worthwhile, it would be a process they would introduce.

**Table 2.** Workshop participants' responses as to whether they routinely checked for open access versions of readings.

| Response | Number of Respondents |
|---|---|
| Always | 1 |
| Sometimes | 6 |
| Occasionally | 6 |
| Never | 4 |

*4.3. RQ3: What Proportion of Content Used under the CLA Licence is Actually Already Available on Open Access, and with an Appropriate Re-Use Licence?*

To ascertain the availability of the sample, searches were made to identify either a pay-per-view or open access version of the article. Of the 296 journal items (188 scanned and 108 digital), 140 (47%) were available on pay per view only and 41 (14%) were not apparently available online at all (see Figure 2). The remaining 113 (38%) were available on open access in some form. This included 33 items (11%) that were available as publisher-hosted Gold open access copies, and 62 items (21%) which were available both on pay-per-view and some other form of open access. The 'Other' open access copies could have

been made available either legally or illegally and are discussed further below. Thus, it would appear that the majority of journal material being copied under the CLA Licence was also available in some other electronic form, which would suggest that the Licence is not being used due to poor availability of digital originals, but perhaps due to convenience, cost or usability. However, the fact that over one-third of content required for teaching purposes was available openly in some form should be of significant interest to HE practitioners.

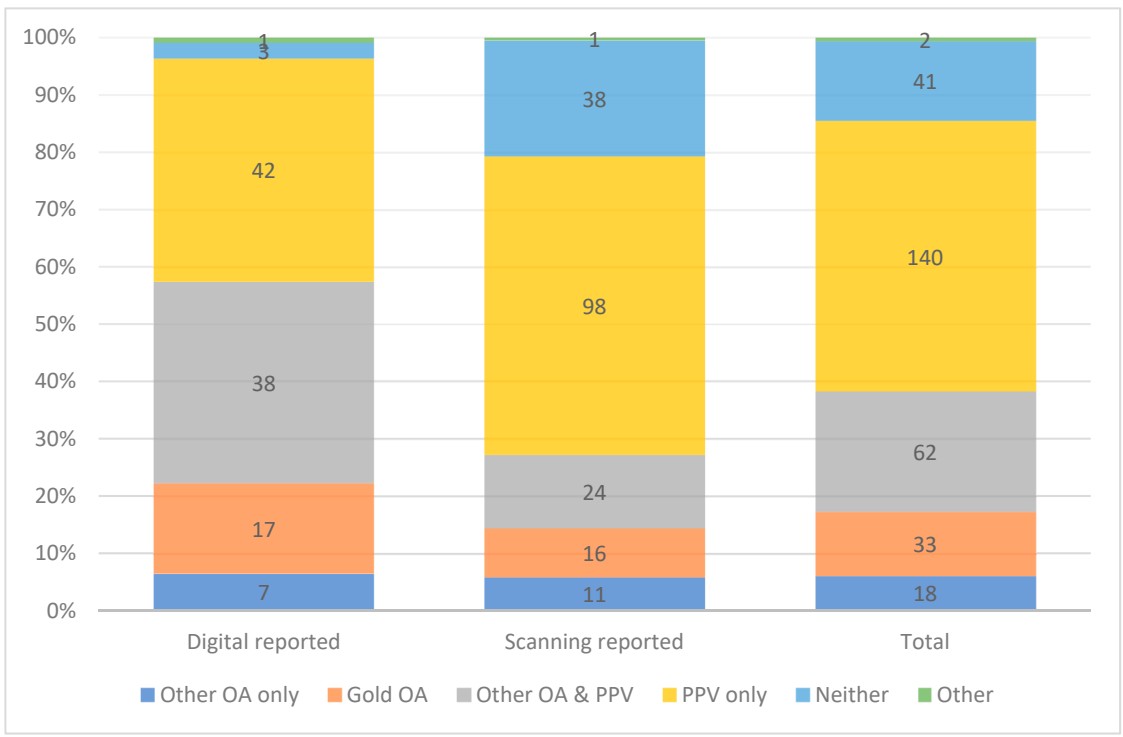

**Figure 2.** Availability of journal article sample via pay-per-view or open access. OA: open access. PPV: pay-per-view.

4.3.1. Open Access Availability of Content.

As outlined in Figure 2, 38% of the sample (113 items) were available on open access. Gold OA copies accounted for 11% of these (33 items) with 'Other' OA copies accounting for the remaining 27% (80 items).

Of the 113 items, OAB correctly identified 34 items (29%), although there were a further 17 cases where it wrongly identified an item as being available on open access when it was not. In such cases, it either took you to an Institutional Repository metadata record wrongly assessing that it contained full-text; to the wrong paper; to a foreign language abstract of the paper; or to a publisher or aggregator site where the full-text was behind a paywall. Unpaywall correctly found 32 items (28%), but with only one false positive. Open Access Button found 4 items not found by Unpaywall, and Unpaywall found 6 items not found by OAB.

Google Scholar found OA copies of the 79 items not found by either Unpaywall or OAB. In many cases Scholar found more than one alternative version of the same item. (see Figure 3). In fact, for 25 of the 79 items, two or more copies could be found. It might be expected that, due to the mission of Unpaywall and Open Access button to seek out only legal OA copies (see Figure 4), that those found only by Google Scholar were not legally available. Whilst this might be true in some cases (see Figure 5), with 23 copies found on ResearchGate and 17 on Academia.edu, it was not always the case. Indeed, of the 79 copies found only by Google Scholar, 18 were available on Gold open access and nine on Institutional Repositories. These analyses would suggest that while Unpaywall and OAB are comparable in terms of their recall of legal open access copies, there remains a proportion of legal

copies they are currently not discovering, and HE practitioners cannot rely on Unpaywall and OAB alone to discover legitimate OA copies.

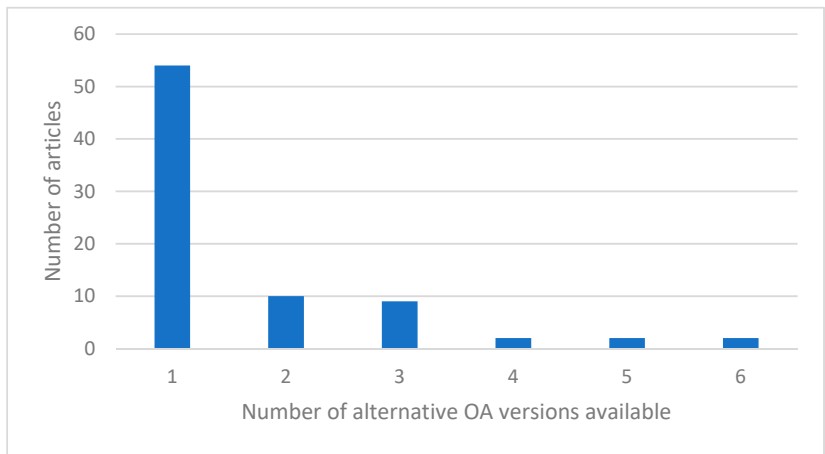

**Figure 3.** Number of alternative OA versions found on Google Scholar.

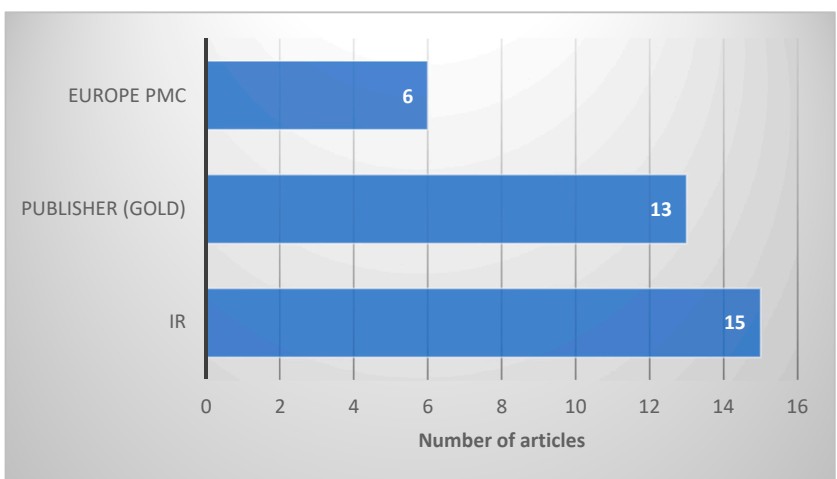

**Figure 4.** Location of copies found via Unpaywall and Open Access Button.

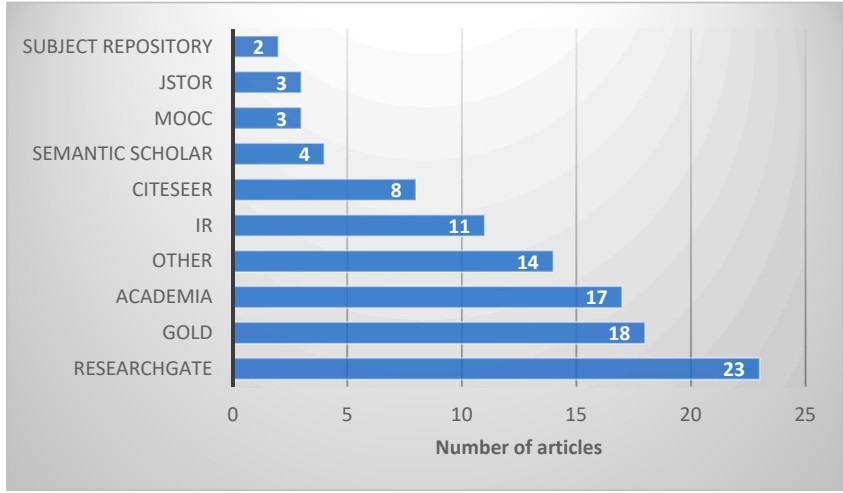

**Figure 5.** Location of all Open Access copies found by Google Scholar.

#### 4.3.2. Licences for OA Material

Of the 113 items available as OA, only 21 (19%) had an associated licence. This was a much lower proportion than the pay-per-view copies (61%). Of this 21, nine were Gold OA items with associated publisher terms and conditions. When analysed, only ten of the 21 OA items with a licence gave clear unequivocal permission for HE libraries to make the content available in an electronic coursepack. Of the remaining eleven items, with four it was likely that the content could be used (the publisher agreement stated that most Gold items were available under a CC-BY licence, but the actual licence was not attached to the paper); with another four it was unclear (it was likely that the copies were illegal, although the licence was permissive), and with the final three it was clear that the content could not be used in this way.

#### 4.4. RQ4: What Percentage of Content Used by UK HEIs under the CLA Licence is Written by UK-Based (or Non-UK-Based) Academics?

By checking institutional affiliation information on the selected journal articles, it was possible to determine that 263 of the 296 items (89%) were written by authors working in academic institutions at the time of publication (see Figure 6). Of the 296, 99 (34%) were written by UK-based academics and 164 (55%) by non-UK-based academics.

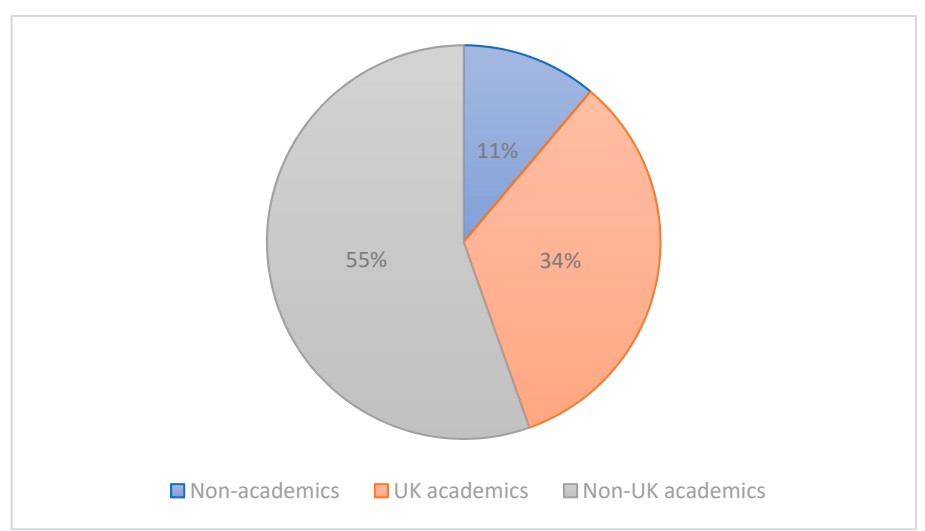

**Figure 6.** Proportion of journal article sample authored by academics.

For comparison purposes, an analysis of the authorship of the 35 most frequently scanned ISBNs was performed. Of the 35 titles, 32 were employed by a university at the time of writing (91%). Twenty-one (60%) were UK academics and eleven (40%) non-UK academics.

### 5. Discussion and Conclusions

This was an interesting analysis and highlights some clear action points for the HE community. The overarching thing to bear in mind is that this analysis only considers the content HEIs felt necessary to scan or digitally copy under the CLA Licence, not the total volume of content that they use in teaching. It is also important to note the large size discrepancies between book content (82%) reused under the licence and journal content (18%). It is very likely that this is in part due to the better suitability of book content (especially textbooks) to support teaching activities, but also due to the general electronic availability of journal articles since about 2000. Indeed about 85% of the digitally copied ISNs were journal articles, whereas 85% of the scanned ISNs were books.

There is no doubt that these sorts of data are extremely valuable to organisations such as Jisc, SCONUL and RLUK who seek to represent the sector and negotiate access to content. It would seem

sensible that these data were simultaneously reported both to the CLA for distribution purposes and to other parties for monitoring purposes in future. It was interesting to observe that not all HEIs supplied author–title information to the CLA as part of their reporting which made it impossible to identify the items being scanned. It is to be hoped that future iterations of the data may be more complete to allow for more accurate analyses.

*5.1. Open Access Availability of Journal Content Being Used to Support Teaching*

Of the journal content copied under the CLA Licence for teaching purposes, 38% (95% CI [32%–44%]) was available on open access in some form, and almost one-third of this was available as Gold OA. This was a particularly interesting finding as previous studies of OA availability have focussed on more recent content. Considering the publication dates of the articles sought spanned almost a century (1918–2017), the fact that such a high proportion was available as OA in some form should be of considerable interest to the UK Library community.

Indeed, taking into consideration the fact that journal content accounted for just 18.5% of all material copied, this suggests that 7% of items (95% CI [1%–13%]) are potentially being copied under the CLA licence unnecessarily. However, interviews showed that seeking out OA copies of journal articles was not routinely performed by HEIs when looking to provide access to this content. This was due to a number of factors including a lack of awareness amongst acquisitions Librarians of the availability of teaching content in this format. There were also some concerns about the legitimacy or permanence of OA content, and the lack of appropriate licences for OA content. The findings suggest that research support teams in university libraries could liaise more closely with the acquisitions staff who were purchasing content for teaching, to help them identify legitimate open access content.

*5.2. Types of OA Content Available*

The different types of OA copies available is a minefield to the inexperienced. A journal article might be a pre-print, an author accepted manuscript or a publisher PDF. Any one of these may be uploaded legally or illegally and may or may not still be there when you next visit that URL. If Librarians are to be persuaded to regularly include searches for OA content in their acquisitions procedures for teaching support, they would benefit from clear guidance as to the best way of approaching this. It was therefore a helpful exercise to test the efficacy of the two main open access content finding services, OAB and Unpaywall, on some 'real-world' high-demand content.

In terms of the percentage of content found, the two were fairly similar, albeit low (28%–29%). However, it was somewhat concerning that OAB found so many false positives (17 of the 113 OA items—15%). Both services found only legally available copies (Gold, IR or PubMed Central). However, it was again interesting to see that they both failed to identify 18 legal Gold OA copies and nine IR copies found by Google Scholar. This exercise would suggest that Unpaywall was the better service of the two for use by UK HE Librarians seeking to find legal content for teaching purposes, but that this should be used in conjunction with Google Scholar or alternative discovery tools, to ensure legitimate OA copies are not missed.

The number of alternative locations available for copies found only by Google Scholar might also be a comfort to Librarians seeking assurances of permanence. Over one-third of the items found by Google Scholar (25 of the 79) provided links to two or more full-text copies. In 21% of cases, three or more copies could be found. Under the 'LOCKSS' principle (Lots of Copies Keeps Stuff Safe) this might provide Librarians with reassurance that should one copy disappear, another may still be available.

*5.3. Available Does Not Mean Re-Usable*

Of course, the fact that an item is available as OA does not mean that it has been mounted legally, nor that it can be legitimately incorporated into an 'electronic coursepack' (a collection of digital outputs for teaching purposes). Although 38% of journal articles were available on open access, a far lower proportion (19% of those that were OA and 7% of the overall sample) came with an associated re-use

licence, and of those that were licensed, even fewer (3% overall) gave clear, unequivocal permission for inclusion in an electronic coursepack.

It is not always clear whether Librarians are legally entitled to draw their students' attention to online content. This is due to the complexity of whether linking is a communication to the public, in addition to the lack of transparency regarding licensing terms. It is true that many countries' copyright laws do provide exceptions that enable Librarians to supply, and students to access copyright material at a time and place convenient to them for research and private study. However, it is known that Librarians are professionally conservative and unlikely to want to take the risk of linking to open access content if there is some doubt as to its legitimacy and the availability of a licence agreement would provide some certainty around their use of the content. Items that can be copied under licence and included in a permanent form through a content delivery system such as a Virtual Learning Environment (VLE) would be preferable to linking to a URL found through Google Scholar. In order to gain the benefits that the mainstream open access movement seeks to provide, it is essential that not only a greater proportion of OA content is clearly licensed, but that those licences are permissive and seen to be reliable by those seeking to build services based on them.

### 5.4. Actual Versus Potential Open Access

The majority of journal content in our sample (89%) was written by academics, although a lower proportion (34%) was written by academics based in the UK. We know from the date range of journal articles copied, that 171 of the 296 journal articles (58%) were written since 2000, when it was certainly technically (if not always legally) possible to make such outputs available on open access. The open access movement is clearly a global movement, focussed on but by no means limited to academia, although with geographical differences in approach. It is somewhat disappointing, therefore, to consider that had all academics retained copyright in their articles [44] and made them available under suitable re-use licences, approximately 58% of the journal content cleared under the CLA Licence need not have been cleared.

### 5.5. Open Access Monographs

A high percentage of the 35 most frequently scanned books were written by academics (91%) and UK academics at that (60%). This raises the question as to how such academics view the re-use of their materials by fellow academics in teaching and how they balance the financial rewards resulting from CLA Licence royalty payments with the reputational and impact rewards of knowing that their works are heavily influencing the next generation through courses of study. If the rewards for them are mainly reputational, then a shift towards open access publishing might be a viable option.

The high percentage of scanned books used in teaching having been written by UK-based academics is of particular interest when considering the UK Research and Innovation announcement around the open access monograph requirement for the next REF [7]. Of course, open access monographs in all fields is a long way off and looks set to be a challenging ride. It is also likely to only affect research monographs rather than textbooks in the first instance, if motivated solely by REF. However, should the community see a move in this direction, it is likely that the value of the CLA Licence to HEIs will reduce in line with an increasing availability of openly available monographs.

### 5.6. Summary

The results of this study indicate for the first time that the state of open access is such that it might provide a viable alternative to copying articles under the CLA Licence to support courses of study. It is hoped that this might encourage university libraries to rethink how they provide access to this content. However, uncertainty around the location, permanence, legality, and licensing of articles may be a cause for concern. Moving forward, there is clearly a role for Open Access advocates and supporters to advise researchers who make content available to do so under a clear re-use licence, and to advise producers of monographs to consider their OA options.

In the interim, the supportive UK copyright communities of practice are well-positioned to offer Librarians support as they seek to explore the use of OA content in teaching. In addition, the production of a guide to locating open access content for use in teaching would be greatly beneficial to the HE community.

**Author Contributions:** Conceptualization, E.G., C.M. and J.S.; Data curation, E.G.; Formal analysis, E.G. and J.S.; Funding acquisition, C.M. and J.S.; Investigation, E.G., C.M., and J.S.; Methodology, E.G., C.M. and J.S.; Project administration, J.S.; Validation, C.M.; Writing—original draft, E.G.; Writing—review and editing, E.G., C.M. and J.S.

**Funding:** This research was funded by Jisc, RLUK, SCONUL and UUK.

**Acknowledgments:** The authors are grateful to the Jisc, RLUK, SCONUL and UUK for funding this research. They are also grateful to Sharon Cocker, Ruth Mallalieu, Neil Sprunt, and Ralph Weedon for assisting with data collection and to the UUK Copyright Negotiating and Advisory Committee for their steer and guidance.

**Conflicts of Interest:** The authors declare no conflict of interest. The funders had no role in the design of the study; in the collection, analyses, or interpretation of data; in the writing of the manuscript, or in the decision to publish the results.

## Appendix A. Interview Questions

Name of Institution:
Name of person being interviewed:
Scanning return data 2017/8:
Total spend on e-resources (from SCONUL data)
Centralised scanning service? Yes/No
Using TADC/DCS/Reporting spreadsheet
Reading list system?
Paper course packs in production?

*Questions*

1. Could you please explain the process by which you acquire/purchase essential readings to support teaching and learning (chapters from books, journal articles)

   ○　Who checks reading lists to advise lecturers on availability?
   ○　What checks do you undertake before deciding to digitise a chapter from a book or an e-journal article?
   ○　Would you check if e-journal articles or chapters are available on open access or electronically?

2. Do you have an e-first policy? Can you tell me more about how that works? What are the terms of this policy?

3. Do you have any problems understanding e-journal licences/Open access terms when sourcing digital content?

   ○　Are there any times you might not rely on this type of content and use the CLA Licence? Why might this be? (Prompt about DRM).
   ○　Which sources would you search to investigate open access content? Who does this type of checking?

4. Could you explain the decision-making process when you rely on the CLA Licence to source content?

   ○　Are there any exceptions to your policy or unusual incidents worthy of mentioning?
   ○　Do you re-check the reading lists on a regular basis?

5.　　Have you observed any changes in the patterns of scanning at your institution in the past X years (increasing/decreasing?)

　　　◦　　Have you any thoughts about what might be leading to these changes

6.　　Have you been audited by the CLA in the last 5 years and if so did it lead to any changes in policy?
7.　　Do you think scanning is being undertaken by staff that is not being reported in your CLA Licence?
8.　　Do you feel the CLA Licence represents good value for money for your institution and why do you say that?
9.　　Is there anything else noteworthy that might inform our research?

**Appendix B. Workshop Questions**

- How do you ensure you purchase information resources needed for teaching and learning purposes?
- Are the licences, models and platforms suitable for teaching purposes?
- What role does the CLA Licence play when sourcing content for teaching and learning purposes?
- Have you observed any changing patterns in relation to your use of the CLA Licence?
- Do you anticipate any changes in your use of the CLA Licence in coming years?
- Is there anything that might lead to any changes in how content is sourced for teaching purposes?
- Do you routinely check for open access versions of readings?
- Do you have a sense that the CLA Licence represents good value for money for your institution?

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
