# Peer review of "The Impact of Open Access on Teaching—How Far Have We Come?"

_publications, doi:10.3390/publications7030056_

Round 1

Reviewer 1 Report

I appreciated having the opportunity to review this important work. I learned some interesting things from your paper and have a few suggestions to help make it even more informative for an even broader audience. My comments and suggestions below follow the outline of your paper.

ABSTRACT

The first 2 sentences are not complete sentences. Please revise.

Line 21, please do not start a sentence with a numeral.

INTRO/LIT

I am education faculty member and researcher who has advocated for OA, but I admit I do not know much about the CLA. Could you provide a little more info, including if it provides services to those outside the UK as well?

Line 42, did you mean “they ALREADY pay to make…”?

Line 55, revise to say “in THE future”.

Line 59, on the previous page you report pounds but here dollars. Why the switch?

Line 76, end with a question mark.

Line 95, do not use & outside parentheses.

Line 103, extra period.

Line 118, who do you mean by “staff”?

Line 135, revise to “However, that aside, it is…”

Line 155, here and elsewhere should be et al. I believe. Also should read “discovered THAT 54.6…”

METHOD

Line 204, this is the first time you use the acronym SCONUL, so please spell it out here.

Line 212, should read “Data were collected…” (also line 224, treat data at plural). And please clarify who it was you were contacting to collect what data in lines 212-13. If the data primarily came from the CLA dataset, who were you interviewing outside of data from describe from NAG?? Please make it clear who was interviewed.

Please provide more info on the type of institutions these were. In the US we have the Carnegie ratings, is there something like that?

Who were the delegates at NAG, university librarians? Please add.

Line 232, so if the record had no ISSN or ISBN, why was it automatically not identifiable? They provided no other info to follow up on, like author and title?? And if an ISSN is so helpful, why do you also need an author name to identify it? Couldn’t you google the rest of the info to find out what the item was? Same with journal name and page numbers…if you have year too, wouldn’t that be enough?

Line 267, please define Gold, Green, and Bronze OA in the lit review.

RESULTS

Line 334, use more speculative language here, like “but PERHAPS due to…”

DISCUSSION

What fields or courses of study were captured in the random sample of articles and chapters? From my own experience I know some fields are more book and monograph dependent and others more journal article dependent due to what is valued for tenure. Is there literature on this? Please add something about this to the intro/lit, results, and discussion. Given the research questions about “courses of study” I kind of expected some results broken down by different fields.

Line 401, “THESE data WERE simultaneously…”

Line 449, what is the equivalent to an “electronic ‘coursepack’”?

As I was reading the last paragraph I wondered about SHERPA/ROMEO…doesn’t this provide good info about our rights and uses of materials from different publishers??

Line 470, it would be helpful here to provide and/or link to some tips for HOW authors cold retain rights.

Lines 481-3, please make clear how books fit in with REF requirements about monographs, as these are not the same, correct?

Last paragraph, it would be amazing to provide a table or tip sheet to folks on how to locate open access content based on your findingsJ

Author Response

Many thanks for taking the time to review our article, please see attachment for point by point responses.  I hope these address all of your helpful comments.

Reviewer 2 Report

This is a really interesting article which adds an important new dimension for considering open access and its impact – often this is solely focussed on the research environment and its researchers, but this research highlights the importance for conceptualising OA as part of the HEI’s wider mission and the potential benefits of OA has on this. It provokes thought by drawing together important threads regarding research, teaching, copyright and university processes, often constrained by siloed working, but which need to be considered as a whole. It also highlights interesting results which further add to existing research regarding copyright and licencing use and understanding, as well as highlighting the unused potential of open access. A great addition to the field.

Overall, this was a well-written, well-structured article. The introduction and literature provided good context for the study and the literature review in particular was well-structured to draw out the multiplicity of many of the themes discussed. The authors delineate clearly between the use and potential for OA for research and OA for teaching. Whilst there are some limitations to this study, they are clearly highlighted where necessary. The over-arching question and four other research questions are well-thought out and are all addressed within the findings and discussion clearly.

The main area for further tweaking is, I believe, the methodology. Whilst well-conceived and the idea of multiple research methods for triangulation is sound, it would improve the understanding of the article as a whole if certain aspects were given further explanation. For example the method section primarily focusses on sampling and does not explain how the data was analysed, if any tools were used etc. For the data collected via interview it would be useful to know who were interviewed – librarians, copyright experts, other HE professionals? Similarly, appending a list of the interview questions/ example of interview transcript/ publishing the data would all improve reproducibility of this study. If possible, quotations from the interviewees in the findings section would help illustrate points clearer. Additionally, it would be useful to highlight when all of the data collection took place, which isn’t detailed for the interviews.

One section in particular which is currently in the findings section (lines 318 – 324) could benefit to moving to the methods section as it relates to the sample size. Additionally some further information could be added to Table 1 to increase clarity of the sample size – for instance a further column could be added for records with author information (lines 246/7) and final sample taken (lines 320-324), this would leave you with one clear table showing exactly how and what the final sample was. It may also help the reader to understand the link between the different data collection methods used, by listing under each of the research questions in the findings section, which of the 3 methods were used.

Some specific comments:

Line 417-19 – I’ve been unable to find the corresponding information within the findings to match the claim in the discussion section – it may be that this refers to information from the literature review, if so this needs to be made explicit, if not further information needs to be added into the findings section if this was expressed by interviewees.

There are some broad generalisations which would hold more weight if they had associated citations, examples included from lines 45-48, 106 and 185.

Line 172 – unaware of the phrase ‘give-away’ nature…a brief explanation would clarify this.

This may be only my personal preference for phraseology but the term ‘on OA’ appears throughout and this phrase may not help with the confusion over open access and how it’s accessed – it makes it sound as there is one single platform and if something is found on it, then it is OA. Things are made available ‘as OA’ or ‘made available openly’ would seem to be a clearer phraseology.

Line 103 – additional full stop.

Line 131 – Stray apostrophe for “call’s”.

The word librarians is sometimes capitalised and sometimes not – lines 135, 138, 426, 450 are examples.

Line 159 – may be helpful to highlight date of study as it was published over 20 years ago.

Line 175 – replace ‘that’ with ‘than’.

Author Response

(The authors gave the same response as above.)

Reviewer 3 Report

Overall: Thank you for this interesting paper, which tackles an engaging and relevant subject and clearly provides some useful revelations. In particular, I would like to commend the introduction and literature review for making a strong and clear case. However, there are some slight mismatches betwixt methods and results that give me slight concerns. I confess to a slight desire to understand more about the qualitative methods employed, given the strength of coverage given to the quantitative ones within the mixed method framework. However, I am confident that the authors will be able to make minor modifications which will enhance the clarity, and I have included some suggestions below. Of all these, I feel the presentation of results is the single area which would benefit significantly from reworking as currently it is the weakest part of the work.

---

Abstract: The second sentence in the abstract runs on at rather an excessive length, and would benefit from being revised into two or three separate ones for clarity. 

42: 'they are to pay' - perhaps 'they are required to pay' might make more sense in context.

54 (and elsewhere): 'text books' - is not 'textbook' the preferred standard? I notice both are used, and standardising on one would be preferable.

58-59: While I appreciate the source is a US one, as the impact references the UK academy, a UK£ value might be desirable.

119-120: The clarity could be improved here, as on first reading it appears to suggest a direct qualitative statement from 45 separate respondents, rather than a qualitative survey output.

151-152: Scientism bias/assumption- scholars in the arts and humanities use research which is decades (and longer). Perhaps tone down the sweeping statement?

214: Listing the researchers' names here is redundant

200-213: This feels less a method and more a statement, as it lacks evidence of the lines of questioning and interview approaches (e.g. structured, reflective, semi-structured, free-form etc) adopted. Providing these would undoubtedly benefit clarity and rigor.

215-220: Again, some value would be added, especially for replication of the work, if the questions used here were appended as a supplementary document or appendix. Nevertheless, this was for me the most interesting part of the study, and I wished equivalent detail or attention had been paid to it in contrast to the data analysis which followed, which seemed to be the heart of the study. I perhaps remain unclear as to whether the workshop was merely a framing exercise, an ancillary piece of work worthy of its own paper or a major research component.

318-366: I found the clarity of what was being revealed, confusing, and despite re-reading this section I remain uncertain of the revelations. The plethora of charts, while providing a much needed break from the numerically rich text, did not aid clarification as much as I would have hoped. Clearly there *are* some very interesting points being made here from the data, but the provision of some plain text summation at the end of these sections would enhance the piece.

391: I'm not sure I'm convinced the analysis has thus far been clearly demonstrated to be 'interesting' (sorry!), perhaps because of the difficulties I found within the results presentation. Related to the results summary suggestion above, perhaps adding 'because...' in the opening line would resolidify the key revelations from the prior section as well as whetting the reader's appetite further.

460 & elsewhere: I have issues with the 'open access movement' and its desires, outcomes and destinations being presented as uniform and homogeneous . As the work of Eve (and others) has demonstrated, it is far (in a best case analysis) more heterogeneous in construction or (worst case) fractured, factionalised and dis-unified. Perhaps phrasing as 'the benefits the mainstream OA movement...' seeks, might be an approach to acknowledge the more radical elements who desire more 'holistically disruptive' impacts to the academic publishing, licensing and access environs than 'professionally conservative' librarians?

465: Suggest rewording '...was written by UK academics' to '...those situated in the UK' to avoid the repeated 'academics' phrase.

Author Response

(The authors gave the same response as above.)
